# Technical note: Reconstructing surface missing aerosol elemental carbon data in long-term series with ensemble learning

Qingxiao Meng[1], Yunjiang Zhang[1*], Sheng Zhong[2], Jie Fang[1], Lili Tang[1], Yongcai Rao[3], Minfeng Zhou[4], Jian Qiu[5], Xiaofeng Xu[6], Jean-Eudes Petit[7], Olivier Favez[8], Xinlei Ge[1]

[1]Collaborative Innovation Center of Atmospheric Environment and Equipment Technology, Jiangsu Key Laboratory of Atmospheric Environment Monitoring and Pollution Control, School of Environmental Science and Engineering, Nanjing University of Information Science and Technology, Nanjing 210044, China

[2]Jiangsu Environmental Monitoring Center, Nanjing, Nanjing 210019, China

[3]Xuzhou Environmental Monitoring Center of Jiangsu, Xuzhou 221018, China

[4]Suzhou Environmental Monitoring Center of Jiangsu, Suzhou 215000, China

[5]Zhenjiang Environmental Monitoring Center of Jiangsu, Zhenjiang 212000, China

[6]Key Laboratory for Aerosol-Cloud-Precipitation of China Meteorological Administration, School of Atmospheric Physics, Nanjing University of Information Science and Technology, Nanjing 210044, China

[7]Laboratoire des Sciences du Climat et de l'Environnement, CEA/Orme des Merisiers, Gif sur Yvette, 91191, France

[8]Institut National de l'Environnement Industriel et des Risques, Verneuil-en-Halatte, F-60550, France

*Correspondence to*: Yunjiang Zhang (yjzhang@nuist.edu.cn)

**Abstract.** Ground-based measurements of elemental carbon (EC) – classified under thermal-optical methods and considered as a surrogate for black carbon – are essential for assessing air quality and evaluating climate impacts. However, data gaps caused by technical challenges impede comprehensive analyses of long-term trends. This study proposed an ensemble learning modeling method to address these challenges. The model used readily accessible ground observation air pollutant data as proxies for EC-related tracers, along with meteorological parameters, to enhance prediction accuracy. It integrated outputs from Gradient Boosting Regression Trees, eXtreme Gradient Boosting, and Random Forest models, combining them through ridge regression to produce robust predictions. We applied this approach to reconstruct hourly EC concentrations from 2013 to 2023 for four cities in Eastern China, filling 45-79% of missing data and improving prediction performance by 8-17% compared to individual models. Over the 11-year period, EC exhibited an overall decline ($-0.20$ to $-0.14$ µg m$^{-3}$ a$^{-1}$), with a more significant decrease from 2013 to 2020 ($-0.24$ to $-0.15$ µg m$^{-3}$ a$^{-1}$). During this time, the average EC concentration in the four cities dropped from 3.26 µg m$^{-3}$ to 1.59 µg m$^{-3}$, followed by a noticeable slowdown in the rate of decline from 2020 to 2023 ($-0.12$ to $-0.04$ µg m$^{-3}$ a$^{-1}$). Additionally, a fixed emission approximation method based on ensemble learning was proposed to quantitatively analyze the drivers of long-term EC trends. The analysis revealed that anthropogenic emission

controls were the predominant contributors, accounting for approximately 92 % of the changes in EC trends from 2013 to 2020. However, their influence weakened post-2020, contributing approximately 80 %. These findings highlighted that while China's Clean Air Actions implemented since 2013 have substantially reduced black carbon concentrations, sustained and enhanced strategies are still necessary to further mitigate black carbon pollution.

## 1 Introduction

Black carbon (BC), derived from the incomplete combustion of fossil or biomass fuels (Bond & Bergstrom, 2006), is a significant component of fine particulate matter ($PM_{2.5}$) in the troposphere. BC has a strong capacity to absorb visible light (Bond et al., 2013), which can directly or indirectly affect global climate change (Ramanathan & Carmichael, 2008) and influence the structure of the urban boundary layer, exacerbating regional air pollution formation (Ding et al., 2016). Additionally, BC particles, which are small in size and rich in toxic substances on their surfaces, pose significant health risks

(Valavanidis et al., 2013), leading to respiratory and cardiovascular diseases (Wei et al., 2023). Therefore, establishing long-term trends of BC is crucial for studying its climate, environmental, and health impacts.

    Methods for determining ground-level BC concentrations and long-term trends generally include ground-based in-situ observations (Peng et al., 2019; Wei et al., 2020), satellite retrievals (Yu et al., 2024; Zhao et al., 2021), and atmospheric chemical transport model simulations (Yang et al., 2021). Satellite-based retrieval methods are often used to characterize

ground-level BC concentrations. For instance, Li et al. (2022) used multi-angle polarization satellite observations to retrieve the spatiotemporal distribution of global BC concentrations. However, satellite data are susceptible to noise originating from clouds and other factors, leading to data gaps and large uncertainties (Bao et al., 2019; Zhao et al., 2021). For example, Gogoi et al. (2023) compared in-situ BC observations in India from 2019 to 2020 with BC concentrations retrieved by the Cloud and Aerosol Imager-2 (CAI-2) onboard the Greenhouse gases Observing Satellite-2 (GOSAT-2) and found a 33% bias. Beyond

measurement uncertainties, satellite retrieval techniques also generally cannot provide long-term, continuous high-time-resolution (e.g., hourly) datasets.

    To address the limitations of satellite retrievals, reanalysis datasets have developed, combining various in-situ, satellite observations, and short-term numerical weather prediction products through data reanalysis techniques. Notable datasets include the ECMWF Re-Analysis Interim (ERA-Interim) (Dee et al., 2011), MODIS Atmosphere and Land Reanalysis

(MODIS) (Levy et al., 2013), and Modern-Era Retrospective Analysis for Research and Applications, version 2 (MERRA-2) (Randles et al., 2017). MERRA-2, in particular, offers convenient high-time-resolution datasets for atmospheric pollutants, including BC (Bali et al., 2017). However, there are uncertainties when comparing ground-based observations to reanalysis data. For instance, several studies found that MERRA-2 overestimated BC concentrations at most Chinese stations by approximately 30% (Ma et al., 2021; Xu et al., 2020; Yu et al., 2024). Atmospheric chemical transport models are also

commonly used to simulate the spatiotemporal distribution of BC concentrations. For example, Matsui (2020) used the Community Atmosphere Model Version 5 (CAM5) and the Aerosol Two-dimensional bin module for foRmation and Aging

Simulation (ATRAS) model to simulate the increase in global BC concentrations from pre-industrial times to the present. However, uncertainties in parameterization schemes for some physical and chemical processes in numerical models remain (Ervens, 2015; Harrison, 2018). Additionally, emission inventories used in models have inherent uncertainties and may not be updated promptly, affecting simulation results (Xu et al., 2021).

In situ observations remain the most direct and effective method for quantifying BC concentrations. In practice, elemental carbon (EC), measured using thermal-optical methods, is often employed as a surrogate for BC (Bond et al., 2013), especially in cases where optical methods are unavailable. These ground-based measurements (EC and/or BC) are central to long-term observational networks worldwide, such as the Atmospheric Science and Chemistry Measurement Network (ASCENT) in the United States (Ng et al., 2022), the Aerosol, Clouds, and Trace Gases Research Infrastructure (ACTRIS) in Europe (Laj et al., 2024), and the China Atmosphere Watch Network (CAWNET) in China (Zhang et al., 2014). While these methods enable the establishment of time series for BC or EC concentrations, they lack the capacity to provide historical data without prior measurements. Furthermore, practical challenges such as instrument malfunctions, routine maintenance, and hardware limitations at observation sites often lead to data gaps. These interruptions hinder the continuity and completeness of long-term datasets, posing significant challenges for trend analysis and comprehensive assessments.

This study introduces an ensemble learning method leveraging ground-based observational data (including in-situ EC and air pollutant measurements), BC column concentration assimilation data, and meteorological datasets. Applying this approach, we successfully reconstructed hourly EC concentration time series for four representative cities in the Yangtze River Delta region in Eastern China from 2013 to 2023. Furthermore, an ensemble learning model was developed to evaluate the drivers of EC trends, enabling a quantitative analysis of the relative contributions of anthropogenic emission reductions and meteorological variations to the EC trends in these cities over the 11-year period.

## 2 Data and Methods

### 2.1 Air pollutant and meteorological data

The observation sites for this study are located in the representative cities of Nanjing, Suzhou, Xuzhou, and Zhenjiang within the Yangtze River Delta city cluster in Eastern China. All these sites are urban monitoring stations, and the coordinates of the sampling sites are provided in Table S1. Gaseous pollutants at the corresponding sites in the four cities, including carbon monoxide (CO), sulfur dioxide ($SO_2$), and nitrogen dioxide ($NO_2$), were sourced from the corresponding sampling sites. The meteorological data used in this study are listed in Table S2. These meteorological conditions are derived from the ERA5 reanalysis dataset from the European Centre for Medium-Range Weather Forecasts (ECMWF). The spatial and temporal resolutions of ERA5 data are $0.25° \times 0.25°$ and 1 hour, respectively. To represent the meteorological conditions at the observation sites, we extracted data from the ECMWF Reanalysis v5 (ERA5) grid cells that correspond to the coordinates of the monitoring stations.

## 2.2 Measurements and inter-comparison of BC and EC

In this study, long-term measurements of EC were conducted across four cities (i.e., Nanjing, Suzhou, Xuzhou, and Zhenjiang), using the Sunset Laboratory semi-continuous OC/EC analyzer (Model-4), which provided hourly time-resolution. The analysis employed the National Institute for Occupational Safety and Health (NIOSH) thermal-optical transmittance (TOT) method to quantify EC. The thermal-optical approach, which forms the basis of the OC/EC analyzer, is detailed extensively in previous studies (Arhami et al., 2006; Birch & Cary, 1996; Jung et al., 2011; Zheng et al., 2014). In addition to EC measurements, refractory black carbon (rBC) data from our previous study in Nanjing (Yang et al., 2019) were used for inter-comparison with EC data. The rBC measurements were obtained using a single-particle soot photometer (SP2) (Liu et al., 2010; Cross et al., 2010). Briefly, the SP2 operates on the principle of laser-induced incandescence (Liu et al., 2010), which involves heating individual rBC particles to high temperatures using a focused laser beam. The incandescence signal emitted during this process is then used to quantify rBC concentration, based on the characteristic heating curve of rBC particles (Liu et al., 2010). Figure S1 shows the relationship between rBC and EC mass concentrations for the Nanjing dataset in this study, along with those from previous work (Pileci et al., 2021). The results revealed a good agreement between rBC and EC (slope = 1.01) in Nanjing, which is consistent with the range reported by Pileci et al. (2021).

## 2.3 BC data from reanalysis or simulation dataset

The Black Carbon Column Mass Density (BCC) and Black Carbon Surface Mass Concentration (BCS) data used in this study were obtained from MERRA-2 (M2T1NXADG, V5.12.4). MERRA-2 is a new reanalysis dataset released by the NASA Global Modeling and Assimilation Office (GMAO) in 2017. The spatial resolution of the data used in this study is $0.5° \times 0.625°$, and the temporal resolution is 1 hour. The BC concentrations in MERRA-2 are estimated by assimilating satellite-derived aerosol optical depth (AOD) into an atmospheric chemical transport model (Gelaro et al., 2017). Since satellite retrievals are based on optical properties, the inferred BC concentrations be affected by optical effects, such as "lensing effect" (Liu et al., 2015), which could potentially lead to an overestimation of BC concentrations. To compare with the simulated BC concentration by atmospheric chemical transport model approach, we used the Tracking Air Pollution in China (TAP) dataset (http://tapdata.org.cn/, last access 20 August 2024) (Geng et al., 2021). In brief, the TAP dataset includes the surface BC concentration data, which is simulated by the community multiscale air quality (CMAQ) model and machine learning method (Geng et al., 2021; Liu et al., 2022). For simplicity, TAP BC did not account for methodological differences between BC and EC measurements (Liu et al., 2022). As a result, these BC data could be influenced by factors such as model assumptions, emission inventories, and meteorological conditions (Yu et al., 2024; Liu et al., 2022).

## 2.4 Ensemble learning model

In this study, we applied the ensemble learning model (EL) approach to address the two major issues. First, we developed a model to reconstruct long-term trend of EC at urban observation sites, filling in missing data. Second, we proposed an

ensemble learning approach to evaluate the driving factors of EC trends, which can quantify the contributions of emission reduction and meteorological variation to the EC trends. The two modelling methodologies integrate the predictions of Gradient Boosting Regression Trees (GBRTs), eXtreme Gradient Boosting (XGBoost), and Random Forest (RF) using ridge regression. GBRTs and XGBoost iteratively train decision tree models respectively, which reduces residuals step by step to make predictions. The final prediction results are weighted sums of the predictions from each tree model, with different weights for each tree. The RF model consists of multiple decision trees, each of which providing a prediction. The prediction method averages the results from each decision tree to obtain the prediction output, with each tree having equal weight.

For multivariate regression analysis, we chose ridge regression over traditional multiple linear regression to account for multicollinearity among the three model outputs. Ridge regression is particularly effective in handling multicollinearity by introducing a regularization term that improves computational stability and reduces the risk of overfitting (Kidwell and Brown, 1982; Hoerl and Kennard, 1970). The final ensemble learning model, integrating the results through ridge regression to determine coefficients ($m_1$, $m_2$ and $m_3$) for each machine learning model, is given by Eq. 1:

$$f_{EL} = m_1 f_{GBRTs} + m_2 f_{XGBoost} + m_3 f_{RF}, \tag{1}$$

## 2.4.1 Reconstructing missing data of EC

Reconstructing long-term EC data involved integrating hourly meteorological variables and emission indicators into an ensemble learning model. Individual ensemble learning models were established for each city. As shown in Table S2, the meteorological variables include 18 factors. The emission indicators include BCC and in-situ surface observations of CO, $SO_2$, and $NO_2$. These air pollutants, along with EC, are mainly associated with fuel combustion processes: CO concentration is closely linked to combustion source activities (such as agricultural crop, forest fires and fossil fuel) (Reid et al., 2005; Wang et al., 2011). $SO_2$ is generally associated with industrial activities (such as coal combustion) (Wang and Chen, 2016), while $NO_2$ primarily originates from vehicle emissions (Carslaw, 2005). All of them could be thereby indirectly indicating EC sources.

In this modeling approach, available in-situ EC observation data for each city, along with corresponding meteorological and emission indicator variables, were used to construct the training and testing datasets. The training set was used to develop the ensemble learning model for data reconstruction, while the testing set served to validate model performance and determine optimal model parameters. Once the model was optimized, all input parameters from 2013 to 2023 were fed into the ensemble learning model to reconstruct long-term EC data. As shown in Figure S2, EC observations in Nanjing included approximately 49000 valid data points, while approximately 90000 points were reconstructed, resulting in a data completion rate of approximately 47%. The completion rates for the other cities were approximately 45% for Suzhou, 50% for Xuzhou, and 79% for Zhenjiang, respectively.

**2.4.2 Driver analysis of long-term trends**

The analysis of drivers behind EC trends differs from the reconstruction of missing EC data by excluding emission indicator variables from the prediction features. To quantify the factors influencing long-term reconstructed EC trends, we developed a machine-learning based fixed emission approximation (FEA) method. Using the reconstructed long-term EC data, we assume each year $i$ ($i = 2013, 2014, ..., 2023$) could serve as a baseline year for the initial anthropogenic emission conditions, with data from the chosen year used to train the model. Reconstructed EC data and meteorological variables from the selected

baseline year are employed as training and testing datasets to build ensemble learning models specific to individual cities. These models are subsequently applied to predict EC concentrations for the entire period from 2013 to 2023. The resulting predictions reflect variations driven solely by meteorological conditions, assuming fixed emissions at baseline-year levels. This approach effectively isolates the contributions of meteorological variations from changes driven by emissions.

The difference in the observed EC concentrations ($\Delta OBS_{(j,k)}$) between two different years ($j, k$) is jointly influenced by

the inter-annual relative changes in EC concentrations driven by meteorological variables ($\Delta MET_{(j,k)}$) and anthropogenic emissions ($\Delta ANT_{(j,k)}$). This relationship is described mathematically in Eq. 2 as follows:

$$\Delta OBS_{(j,k)} = \Delta ANT_{(j,k)} + \Delta MET_{(j,k)}, \tag{2}$$

This equation (Eq. 2) assumes that changes in meteorology and emissions are the primary factors affecting the observed EC concentration variations between years, allowing for a decomposition of their respective contributions to the trends. The

$\Delta MET_{(j,k)}$ can be estimated by comparing the predicted EC concentrations for two different years ($C_{MET(i,j)}$ and $C_{MET(i,k)}$), where j and k represent the years for which predictions are made using the model trained with data from baseline year $i$. Specifically, $j = 2013, 2014, ..., 2022, k = 2014, 2015, ..., 2023$, and $k > j$). This relationship is expressed in Eq. 3, which isolates the influence of meteorological variations on inter-annual differences in EC concentrations, assuming emissions at the baseline year $i$ level.

$$\Delta MET_{(j,k)} = C_{MET(i,k)} - C_{MET(i,j)}, \tag{3}$$

Thus, the impact of anthropogenic emission controls on changes in EC concentrations can be determined by subtracting $\Delta MET_{(j,k)}$, which reflects the contribution of meteorological variations, from $\Delta OBS_{(j,k)}$, the observed inter-annual EC concentration change. This relationship is expressed in Eq. 4:

$$\Delta ANT_{(j,k)} = \Delta OBS_{(j,k)} - \Delta MET_{(j,k)} \tag{4}$$

Here, $\Delta ANT_{(j,k)}$ represents the portion of the EC concentration change attributable to anthropogenic emission controls during the period between years $j$ and $k$.

Based on the FEA method, we further proposed an analytical approach to assess the impact of short-term pandemic lockdowns. Specifically, we analysed the relative contribution of the COVID-19 lockdown (LD) period, which lasted from

January 26 to February 17, 2020 (Huang et al., 2021). The calculation of the impact of the 2020 LD on pollutant concentrations is determined by the proportion of the LD period samples within the entire year of 2020 ($r_{LD}$), as well as the difference between observed values ($OBS_{LD}$) and predicted values ($C_{MET(i,LD)}$) during the pandemic period. This allows us to calculate the total contribution of the COVID-19 lockdown and the emission control effects between 2020 and the baseline modeling year on pollutants. Assuming that the contribution of anthropogenic emissions to pollutants during the winter of 2019 and other years remains stable, this can be calculated by the difference between observed values ($OBS_{NLD}$) and predicted values ($C_{MET(i,NLD)}$) during the remaining time in 2020. Therefore, the contribution of the 2020 LD to pollutants is represented by Eq. 5:

$$COVID-19_{(i,LD)} = r_{LD}\big(OBS_{LD} - C_{MET(i,LD)}\big) - r_{LD}\big(OBS_{NLD} - C_{MET(i,NLD)}\big) \tag{5}$$

### 2.4.3 Model evaluation and uncertainty analysis

In this study, the model performance was assessed by comparing the reconstructed results for the test set with observational data. To evaluate the deviation between the observed ground-based values ($y_{obs}$) and the reconstructed predicted values ($y_{pre}$), we used several statistical metrics: root mean square error (RMSE), mean squared error (MSE), mean absolute error (MAE), and the correlation coefficient (R), as defined by the following Eqs 6, 7, and 8:

$$RMSE = \sqrt{\frac{1}{n}\sum_{i=1}^{n}\big(y_{obs} - y_{pre}\big)^2} \, , \tag{6}$$

$$MSE = \frac{1}{n}\sum_{i=1}^{n}\big(y_{obs} - y_{pre}\big)^2 \, , \tag{7}$$

$$MAE = \frac{1}{n}\sum_{i=1}^{n}\big|y_{obs} - y_{pre}\big| \, , \tag{8}$$

To validate and assess the model's performance in reconstructing missing data, we evaluated the prediction error metrics on the test set, including RMSE, MSE, MAE, and R. Figures 1 and S3 summarize the performance metrics for individual models, including XGBoost ($R=0.81\pm0.05$), GBRTs ($R=0.84\pm0.03$), and RF ($R=0.87\pm0.03$), as well as for the ensemble learning model. The ensemble machine learning model exhibited a notable improvement, achieving $R=0.94\pm0.01$, which represents a 17% enhancement over the XGBoost model. Additionally, MAE, MSE, and RMSE decreased by 47%, 67%, and 42%, respectively. These results highlight the superior performance of the ensemble machine learning model, showcasing its robustness and accuracy in reconstructing missing data. As shown in Table S3, the ensemble model also yielded the best performance within the FEA framework. We further evaluated the importance of the MERRA-2 black carbon concentration (BCC) as a predictor by testing the model's performance both with and without this variable (see Figure S4). Inclusion of MERRA-2 BCC significantly improved the model's performance across all evaluation metrics, confirming it as a key contributor to model accuracy.

To further assess the influence of emission changes and test model robustness, we trained an alternative version of the model excluding all emission indicator variables. Compared to the full model, this version showed a marked degradation in performance (see Figure S5). This result highlights the critical role of including emission-related variables for ensuring accurate and reliable predictions. Additionally, as illustrated in Figure S6, the CO/NO$_2$ ratio during periods with valid EC

observations closely resembled that during EC-missing periods. This consistency suggests that interannual variation in these emission indicator ratios likely exerted minimal influence on the model's predictive performance. Finally, long-term trends in monthly mean EC or BC concentrations were assessed using the non-parametric Mann-Kendall (MK) trend test. To account for seasonal variability, trend slopes were derived using the seasonal Theil-Sen estimator, enhancing robustness in the presence of periodic fluctuations.

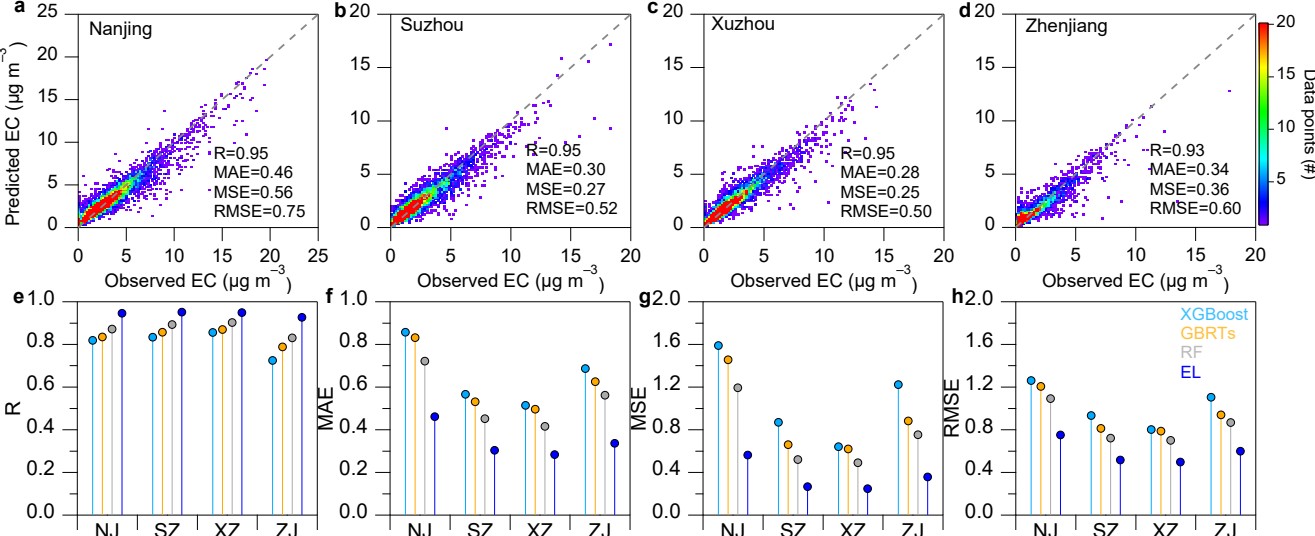

**Figure 1.** Performance evaluation of the models for reconstructing hourly EC concentration. **a-d** Comparison of the ensemble learning model predicted concentration and observed EC concentration. The legend indicates the number of data points at each binning interval, which is approximately 0.25 µg m⁻³. **e-f** Comparison of the model performance parameters for four cities, i.e., Nanjing (NJ), Suzhou (SZ), Xuzhou (XZ), Zhenjiang (ZJ).

To evaluate the relative difference in the analysis of EC trend change drivers, we proposed a method for quantifying the relative difference in the FEA. The term $\Delta ANT_{(i,j)}$ is determined by subtracting $\Delta MET_{(i,j)}$ from $\Delta OBS_{(i,j)}$. By substituting Eq. 3 from the manuscript into this calculation, the relationship is expressed as shown in Eq. 9. The term $C_{MET(i,i)}$ was the self-prediction for the year $i$ based on the training year $i$. This formulation allows for a systematic evaluation of the uncertainties associated with the FEA approach, ensuring robust attribution of trends to anthropogenic emission controls and meteorological variations. Similarly, by substituting results for the two years $i$ and $j$, we can obtain $\Delta ANT_{(j,i)}$, as shown in Eq. 10.

$$\Delta ANT_{(i,j)} = \Delta OBS_{(i,j)} - (C_{MET(i,j)} - C_{MET(i,i)}) , \tag{9}$$

$$\Delta ANT_{(j,i)} = \Delta OBS_{(j,i)} - (C_{MET(j,i)} - C_{MET(j,j)}) , \tag{10}$$

If the FEA method was entirely free of relative difference, the relationship $(C_{MET(i,j)} - C_{MET(i,i)}) + (C_{MET(j,i)} - C_{MET(j,j)}) = 0$ would hold true, implying that $\Delta ANT_{(i,j)} + \Delta ANT_{(j,i)} = 0$. However, as with any method, some degree of

relative difference could be unavoidable. To account for this, the relative difference of the data for year $j$, predicted using year $i$ as the training data ($(Y_{i(j)})$), was determined by the absolute value of the sum of $\Delta ANT_{(i,j)}$ and $\Delta ANT_{(j,i)}$, and then normalizing by $C_{MET(i,j)}$. This calculation is expressed in Eq. 11:

$$Y_{i(j)} = \frac{|\Delta ANT_{(i,j)}+\Delta ANT_{(j,i)}|}{C_{MET(i,j)}}, \tag{11}$$

This approach provided a quantifiable measure of relative difference inherent in the FEA method, facilitating a more robust evaluation of the predictions. When $i = j$, $\Delta ANT_{(i,j)} = \Delta ANT_{(j,i)} = 0$, and the relative difference calculation at this point is expressed by Eq. 12.

$$Y_{i(i)} = \frac{|OBS_i - C_{MET(i,i)}|}{C_{MET(i,i)}}, \tag{12}$$

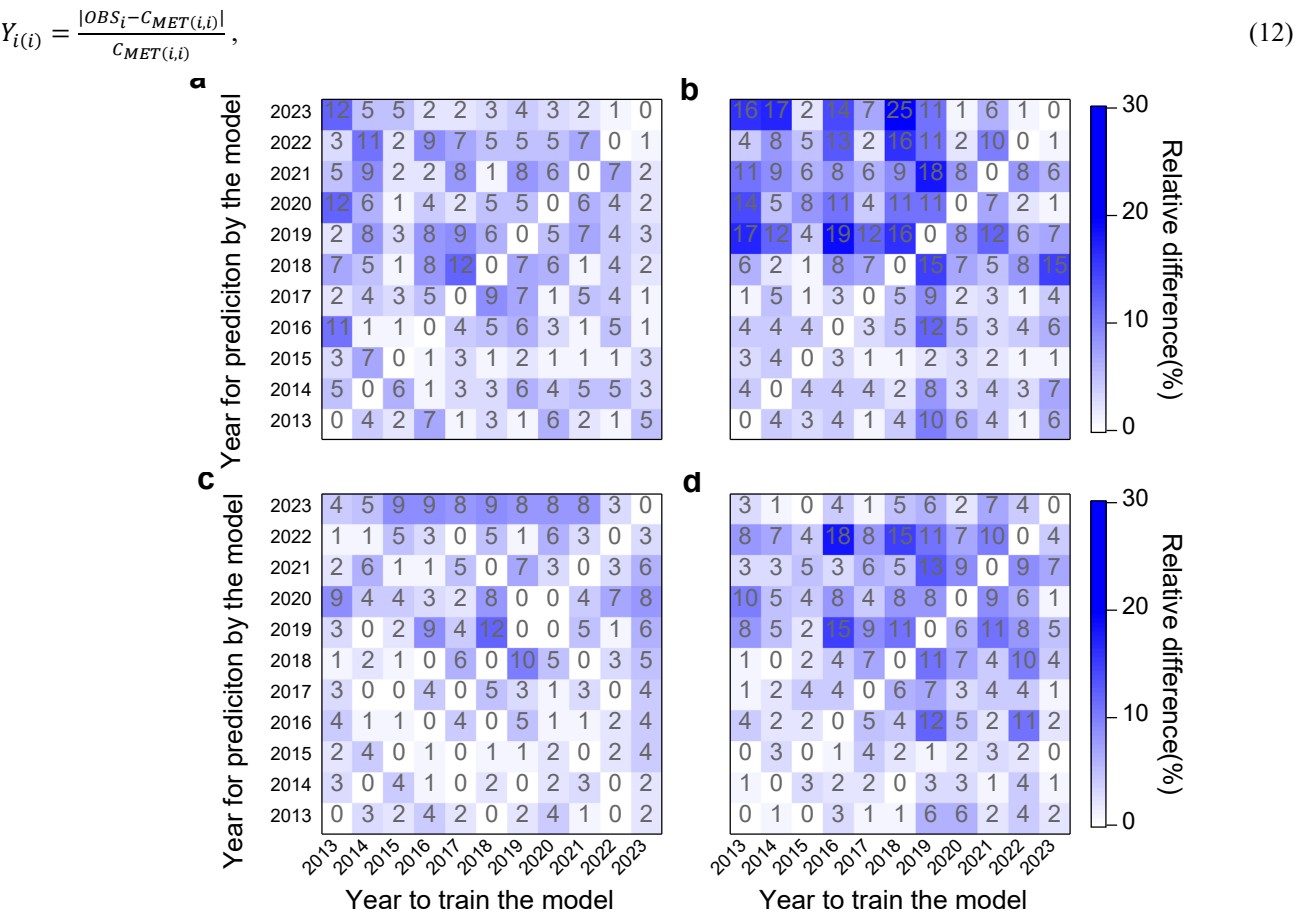

**Figure 2.** Cross-matrix relative difference analysis of the FEA method (a. Nanjing, b. Suzhou, c. Xuzhou, and d. Yangzhou). The relative difference here refers to the relative difference of the results obtained from different year to train the model and different year for prediction by the model.

    As shown in Figure 2, the average relative difference across the four cities over the eleven-year period is approximately 250    4 ± 4%. The lowest relative difference was observed in Xuzhou, with 3 ± 3%, while the highest was in Suzhou, with 6 ± 5%.

Because this study incorporates reconstructed EC data, potential discrepancies between model-derived training data and actual observations may introduce additional uncertainty into the results. To assess this, we selected Nanjing as a representative test case. Specifically, we reconstructed the 2013–2023 EC dataset using training data from two distinct time periods: 2013–2020 and 2014–2019. We then applied the FEA method to both reconstructions and compared the outcomes. As shown in Figure S7, while some variations exist between the two sets of results, the average difference remains within approximately 10%. This finding suggests that the choice of training dataset can introduce a moderate degree of uncertainty to the FEA results— an inherent characteristic of ensemble learning and other statistical modeling approaches. Nonetheless, the relatively small magnitude of this difference reinforces the reliability and generalizability of the machine-learning-based FEA framework. In Section 3.2, we further compare the FEA method with the widely used de-weathered approach proposed by Grange et al. (2018), to evaluate the consistency and applicability of different trend attribution frameworks.

## 3 Results and discussion

### 3.1 Reconstruction of missing data of EC and comparison

As illustrated in Figure 3a–d, the reconstructed EC concentrations in all four cities closely track the trends observed in the ground-based measurements. These reconstructed values also exhibit strong agreement with the temporal patterns of TAP-derived BC concentrations. Over the 11-year study period, both reconstructed EC and TAP BC showed statistically significant and consistent decreasing trends, with rates ranging from –0.20 to –0.14 $\mu g\ m^{-3}\ a^{-1}$ and –0.22 to –0.14 $\mu g\ m^{-3}\ a^{-1}$, respectively ($P < 0.05$). In contrast, the MERRA-2 BC product displays weaker or non-significant declining trends, particularly in Suzhou and Xuzhou, where the trend was not statistically significant ($P > 0.05$). To investigate this inconsistency, we applied the FEA method to isolate the meteorological contributions to the observed trends in MERRA-2 BC. As shown in Figure S8, MERRA-2 BC trends exhibit good agreement with the meteorological-driven BC trend derived from the FEA method, suggesting that the interannual variability in MERRA-2 BC was largely driven by meteorological factors. In contrast, the TAP BC dataset showed clear downward trends that were more closely aligned with the changes observed in our reconstructed EC data (Figures 3b and 3c). This indicated that the TAP dataset for the two cities was more sensitive to the anthropogenic emission reductions and better captured their influence on BC concentration trends in recent years. Additionally, MERRA-2 BC consistently overestimates EC concentrations compared to both ground-based measurements and reconstructed data. This overestimation has also been reported in previous studies (e.g., Xu et al., 2020), which documented substantial overpredictions of MERRA-2 BC in urban areas in the Yangtze River Delta of Eastern China, such as Shanghai and Hangzhou.

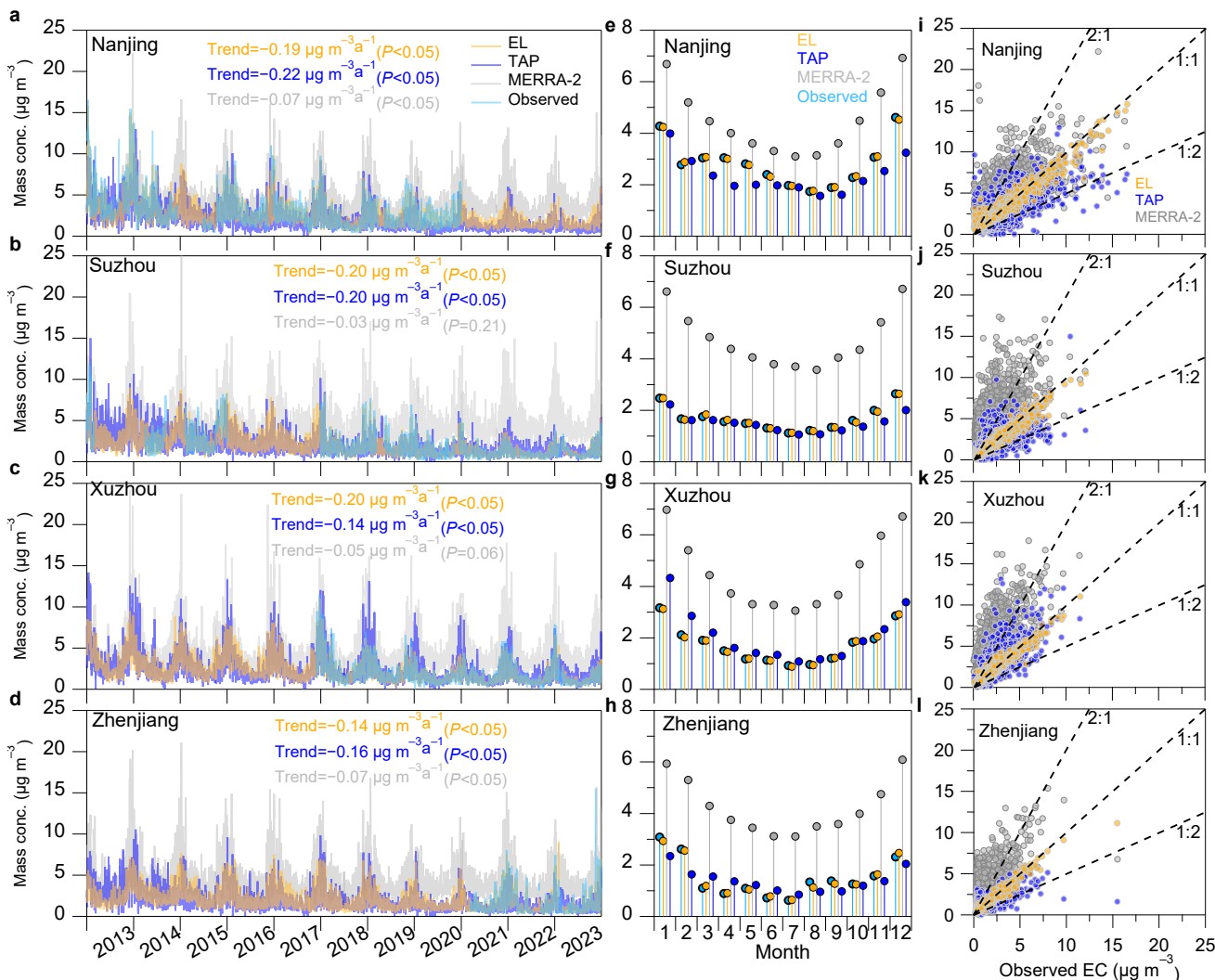

**Figure 3.** Comparison of EC or BC concentration from different data sets. **a-d** Trends in EC or BC concentration from the four data sets (EL, MERRA-2, TAP, and observation). **e-f** Monthly variations. **i-l** Relationship of reconstructed, MERRA-2, and TAP modelled EC or BC with observed EC.

Figure 3e–h further demonstrates that the reconstructed EC data capture monthly variations with high fidelity relative to in-situ observations. All four datasets exhibit a pronounced seasonal cycle, characterized by elevated concentrations in autumn and winter and lower levels in spring and summer. This seasonal pattern highlights the need for more stringent BC emission controls during the cold months. Among the datasets, TAP-derived BC closely tracks the monthly variability observed in ground-based measurements, whereas the MERRA-2 BC significantly overestimates concentrations—by approximately 147% on average. The discrepancies in MERRA-2 estimates likely arise from two key factors. First, the relatively coarse spatial

resolution of MERRA-2 ($0.5° × 0.625°$; Randles et al., 2017) means that a grid cell may span diverse land-use types—including urban, suburban, and background regions—thereby smoothing out localized pollution signals and/or inflating background contributions. Second, uncertainties associated with satellite data assimilation—particularly under conditions of severe pollution, cloud cover, or precipitation—may introduce biases in MERRA-2 estimates (Xu et al., 2020), thereby limiting their accuracy in reproducing station-scale EC variability. As shown in Figures 3i–l and detailed in Table S4, MERRA-2 BC still maintains a moderate correlation with in-situ observations ($R = 0.65 ± 0.05$), despite its substantial overestimation. In contrast, the TAP BC exhibited better agreement with ground-based data ($R = 0.69 ± 0.04$). The highest correspondence, however, was observed for the reconstructed dataset, which achieved an exceptional correlation coefficient of $R = 0.97$. This result could highlight the high fidelity and robustness of the reconstruction approach during periods with available observations.

To assess the quality of the reconstructed missing data, we analyzed its correlation with co-located air pollutants (CO and $NO_2$). As shown in Figure S9, the observed EC concentrations exhibit strong correlations with CO ($R = 0.66 ± 0.10$) and $NO_2$ ($R = 0.71 ± 0.06$) over the entire study period. Similarly, the reconstructed EC concentrations for the missing data demonstrate even better correlations, with $R$ values of $0.80 ± 0.06$ for CO and $0.85 ± 0.04$ for $NO_2$ (Figure S10a-h), respectively. Furthermore, the correlation between the TAP BC and the observed EC concentrations ($R = 0.65 ± 0.05$) was stronger than that observed for MERRA-2. As shown in Figure S10m-p, the reconstructed EC concentrations for the missing data also exhibit a good correlation ($R = 0.72 ± 0.04$) with the TAP BC, with an approximately 11% difference between the two datasets. The slope between TAP and observed EC was $0.60 ± 0.20$, while the slope between TAP and the reconstructed EC was $0.68 ± 0.06$ (see Table S4). These comparisons suggest that the reconstructed data was reasonably accurate when compared to ground-based observations.

Using the reconstructed EC data, we analyzed the trends in EC concentrations across the four cities from 2013 to 2023 (see Table S5). The EC concentrations have significantly decreased, with reductions of 61% in Nanjing and Suzhou, 59% in Xuzhou, and 47% in Zhenjiang compared to 2013 levels. Specifically, in Nanjing and Xuzhou, the annual average EC concentrations decreased by $2.60\ \mu g\ m^{-3}$ and $1.88\ \mu g\ m^{-3}$, respectively, over the 11-year period. This trend aligns with findings by Zhou et al. (2024), who reported similar reductions from 2013 to 2020 based on observational data. These reductions highlighted the effectiveness of China's emission reduction policies in mitigating BC in the Yangtze River Delta. In Suzhou and Zhenjiang, EC concentrations decreased by $1.84\ \mu g\ m^{-3}$ and $1.17\ \mu g\ m^{-3}$, respectively. The average EC concentrations over the 11-year period were $2.01 ± 0.77\ \mu g\ m^{-3}$ in Suzhou and $1.98 ± 0.50\ \mu g\ m^{-3}$ in Zhenjiang, which were significantly lower than those in the more industrialized cities of Nanjing and Xuzhou.

As shown in Figures S11a-d, the diurnal variations in EC concentrations in the four cities have significantly decreased over the 11-year period, with peaks during morning and evening rush hours, primarily due to vehicle emissions. Notably, the rate of EC reduction from 2020 to 2023 ($-0.12$ to $-0.04\ \mu g\ m^{-3}\ a^{-1}$) was significantly lower than the overall rate from 2013 to 2023 ($-0.20$ to $-0.14\ \mu g\ m^{-3}\ a^{-1}$). Zhao et al. (2024) observed a slowdown in the decline of $CO/CO_2$ concentrations in the Yangtze River Delta, attributing this trend to limited improvements in the combustion efficiency of anthropogenic sources in recent years. Figures 3a-d and S11a-d further reveal that EC concentrations in the four cities were substantially lower during

the COVID-19 pandemic (2020–2022). This aligns with findings by Cui et al. (2021), who reported significant reductions in BC concentrations during the pandemic due to lockdown measures. However, EC concentrations in Nanjing and Suzhou rebounded in 2023, likely due to a resurgence of anthropogenic emissions. Similarly, Liu et al. (2024) observed that global carbon emissions, which temporarily decreased during the pandemic in 2021, rebounded and exceeded previous levels in 2022 and 2023. Despite the significant reductions in EC concentrations across the Yangtze River Delta over the past 11 years, the rate of decrease has slowed in recent years.

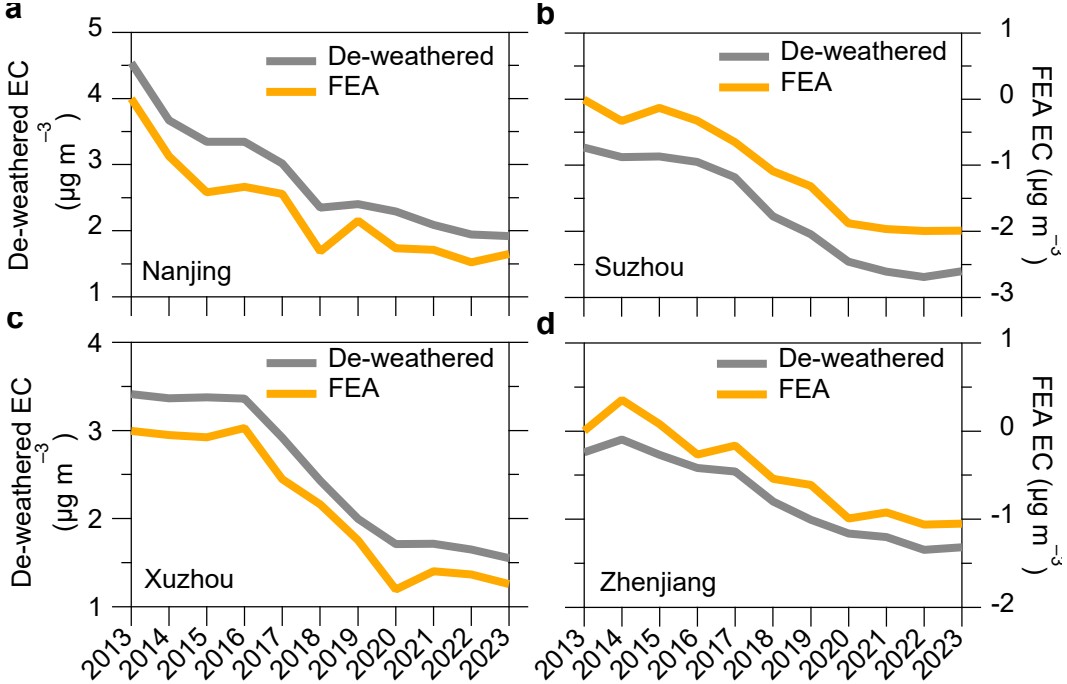

**Figure 4.** Trend in emission-driven EC from 2013 to 2023. Comparison of the results obtained from two different methods, including the FEA method developed in this study and the widely used de-weathered method.

### 3.2 Drivers of the EC trends

To disentangle the effects of meteorology and anthropogenic emissions on EC trends, we also applied a machine-learning-based meteorological normalization method, also known as the de-weathered method (Grange et al., 2018). This method has been widely used to assess the drivers of trends in air pollutants and aerosol chemical composition (Li et al., 2023; Vu et al., 2019; Zhang et al., 2019b; Zhou et al., 2022). The normalization approach developed by Grange et al. (2018) adjusts pollutant concentrations by removing the influence of meteorological variability, thereby isolating the effects of emission control measures. Specifically, the method builds a statistical model to quantify the relationship between pollutant concentrations and meteorological and temporal variables. It then performs 1,000 resamplings of historical meteorological data while holding

temporal variables constant at their observed values. The model calculates pollutant concentrations across all resampled meteorological scenarios, and the ensemble average of these predictions represents the meteorologically normalized (i.e., de-weathered) concentration. This process enables a more accurate attribution of observed trends to changes in emissions. Detailed methodology can be found in Grange et al. (2018) and related studies (Vu et al., 2019; Zhang et al., 2019b; Zhou et al., 2022). This method can effectively separate the impacts of emissions and meteorology on these trends. Detailed methodologies of this meteorological normalization can be found in previous studies (e.g., Grange et al., 2018; Vu et al., 2019; Zhang et al., 2019b; Zhou et al., 2022). To further validate our proposed FEA method, we compared its results against those from the traditional meteorological normalization approach. As shown in Figure 4, both methods yielded highly consistent anthropogenic emission-driven trends in EC across the four cities. Additionally, Table S6 shows that all emission-driven trends derived from both methods passed the Mann-Kendall test ($P < 0.05$), with slope differences of less than approximately 8%. The discrepancies between the two methods remained within approximately 10%, demonstrating strong agreement and supporting the robustness of the emission-driven trends derived from our FEA method.

To quantify the driving factors behind the observed trends in elemental carbon (EC)—specifically anthropogenic emissions and meteorological variations—we conducted a comprehensive attribution analysis. As shown in Figure S12, EC concentrations exhibited a significant decreasing trend from 2013 to 2023. Meteorological factors played only a minor role, contributing $9 \pm 1\%$ to the observed decline, while anthropogenic emission control measures accounted for the remaining $91 \pm 1\%$. As illustrated in Figure S13, we estimated the meteorology-driven component of EC trends using both the conventional de-weathering method and our proposed FEA method. The results from both approaches indicated that meteorology-driven EC trends remained relatively flat over the 11-year period, further confirming that interannual variations in meteorological conditions had limited impact on the long-term EC trend. This also reinforces the reliability of the FEA method. Table S7 presents additional evidence: concentrations of co-emitted pollutants such as CO, $SO_2$, and $NO_2$ also declined significantly across all four cities from 2013 to 2023, corroborating that the downward trend in EC was predominantly driven by emission control efforts rather than meteorological variability. Diurnal patterns of EC, shown in Figure S11a–d, consistently revealed peak concentrations during morning and evening rush hours. However, the magnitude of these peaks has declined markedly in recent years. Correspondingly, Figure S11e–h shows the diurnal profile of emission control-driven EC relative to 2013, highlighting substantial reductions during rush-hour periods—a clear indication of the effectiveness of vehicle emission control policies.

Figure S6a–d and Table S7 further demonstrate strong correlations between EC and $NO_2$ concentrations, suggesting that vehicle emissions are a major contributor to urban BC levels. The observed reductions in $NO_2$ in all four cities underscore the impact of transportation-related emission control measures on declining EC concentrations. We further examined rush-hour EC trends in Figures S14 and S15 and found more pronounced declines during these periods compared to the interannual average, indicating that targeted control strategies for mobile sources have been particularly effective. This finding aligns with previous work (e.g., Zheng et al., 2018), which reported substantial reductions in BC emissions from the transportation sector in China between 2010 and 2017.

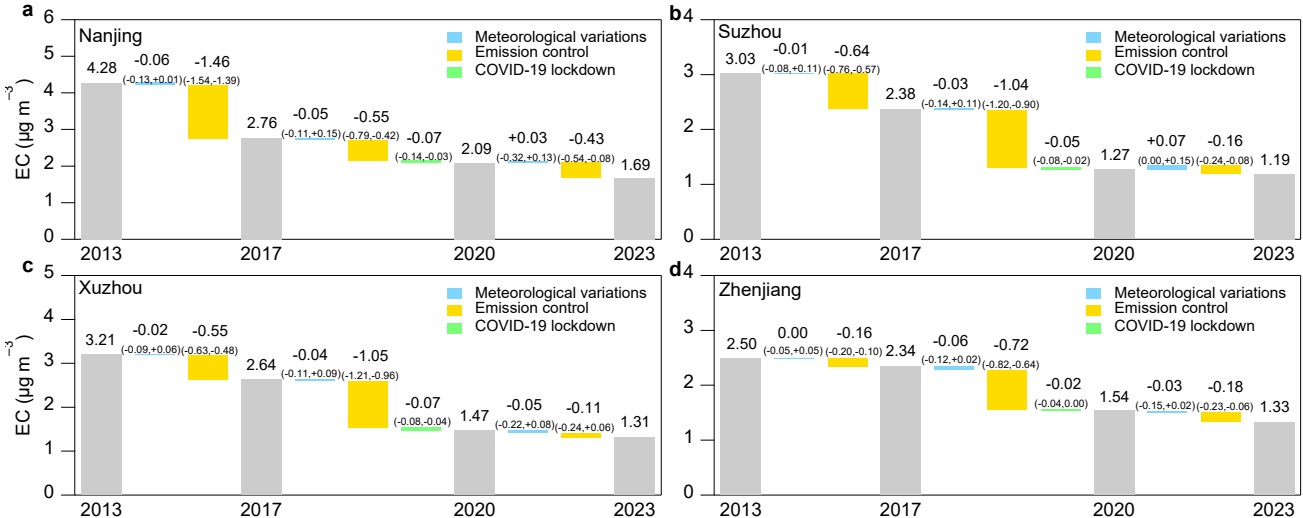

**Figure 5.** Drivers of the EC trend from 2013 to 2023. a-d the contributions of anthropogenic emission control, meteorological variations and COVID-19 lockdown on the trends in EC concentration in the four cities (The values in parentheses represent the minimum to maximum values of the driving contributions analyzed by the FEA method.).

Following the launch of China's Air Pollution Prevention and Control Action Plan in 2013, significant advances in air quality management were achieved. For example, Zhang et al. (2019a) attributed substantial decreases in $PM_{2.5}$ concentrations between 2013 and 2017 mainly to reductions in anthropogenic emissions. To assess the impact of different policy phases (Geng et al., 2024; Zhang et al., 2019a), we divided the 11-year study period into three stages: 2013–2017, 2018–2020, and 2021–2023 (see Figure 5). During the first stage (2013–2017), cities in the Yangtze River Delta implemented targeted control strategies. Nanjing and Suzhou, as economically developed industrial centers, prioritized the management of industrial parks and vehicle emission standards. Xuzhou, dominated by coal and heavy industries, emphasized curbing coal consumption and industrial emissions (Guo et al., 2022; Zhang et al., 2018). These coordinated efforts led to considerable reductions in EC: concentrations decreased by 36% in Nanjing, 21% in Suzhou, 18% in Xuzhou, and 6% in Zhenjiang. The smaller reduction observed in Zhenjiang may be attributed to its lower initial EC levels, limiting the absolute effect of control measures. Nonetheless, anthropogenic emissions remained the dominant driver of EC reductions, contributing 96%, 99%, 96%, and 98% of the observed changes in the four cities, respectively. During the second stage (2018–2020), further progress was achieved under the Three-Year Action Plan for Clean Air. EC concentrations declined by 24% (Nanjing), 47% (Suzhou), 44% (Xuzhou), and 34% (Zhenjiang). Average EC reductions attributed to anthropogenic emissions were −0.82 μg m⁻³ in Nanjing and −1.25 μg m⁻³ in Xuzhou. These reductions were likely amplified by the COVID-19 lockdowns, which significantly curtailed industrial activities and transportation in these cities. As shown in Figure 5, we quantified the impact of the lockdown on EC levels, estimating that the pandemic contributed 3–10% of the total EC reduction from 2017 to 2020—an effect stemming from both emission reductions and changes in human activity. This is consistent with Zheng et al. (2021), who reported large-

scale BC emission reductions during the lockdowns of early 2020, especially from industrial, residential, and transportation sources. In the third stage (2021–2023), the rate of EC decline began to slow. Reductions were 20% in Nanjing, 7% in Suzhou, 11% in Xuzhou, and 14% in Zhenjiang, primarily due to diminishing marginal returns from existing emission control measures. Meteorological influences during this period remained consistent with their contribution over the full 11-year span. Nevertheless, anthropogenic emissions still accounted for 93% (Nanjing), 68% (Suzhou), 67% (Xuzhou), and 86% (Zhenjiang) of the EC decline from 2020 to 2023.

## 4 Conclusion and implication

This study developed an ensemble learning approach to reconstruct hourly long-term in-situ elemental carbon (EC) data using meteorological parameters and co-emitted source indicators. Applied to four cities in the Yangtze River Delta (YRD) region of eastern China, the ensemble model demonstrated superior performance compared to individual algorithms—outperforming XGBoost, GBRT, and Random Forest by 17%, 13%, and 8%, respectively. On average, the reconstructed dataset successfully filled 55% of the missing EC observations. Its reliability was validated through consistency with ground-based EC, CO, and $NO_2$ measurements, as well as with TAP BC observations. The proposed approach provides an efficient and scalable solution to address missing EC data in long-term observational records. It also offers a means to reduce uncertainties in satellite-derived and reanalysis-based BC datasets. Over the 11-year study period, EC concentrations exhibited an overall declining trend, ranging from –0.20 to –0.14 µg m$^{-3}$ a$^{-1}$. A more rapid decline was observed during 2013–2020 (–0.24 to –0.15 µg m$^{-3}$ a$^{-1}$), followed by a notable deceleration between 2020 and 2023 (–0.12 to –0.04 µg m$^{-3}$ a$^{-1}$).

To disentangle the drivers of EC trends, we developed the FEA method based on the ensemble model to quantify the respective contributions of anthropogenic emissions and meteorological variability. The results indicate that emission controls accounted for 91% of the overall EC decline from 2013 to 2023, underscoring their dominant role. The slowed decline during 2020–2023 was mainly attributed to weakened contributions from anthropogenic emission reductions, which ranged from 67% to 93% across the four cities. The ensemble learning framework proposed in this study offers a robust and generalizable tool for reconstructing incomplete air quality datasets, with potential applications to a broad range of atmospheric pollutants. Furthermore, the integration of machine-learning-based reconstruction with the FEA method presented an effective approach for trend attribution—particularly for long-term observational records. While this method has proven effective for trend analysis of primary particulate matter, such as EC, in this study, its applicability to other atmospheric constituents with different sources and data characteristics remains to be evaluated. Future work could explore its performance and quantify associated uncertainties across diverse species and datasets.

Overall, our findings highlight the substantial impact of emission control policies on mitigating urban particulate EC pollution in the YRD. To sustain and accelerate reductions in EC and broader black carbon concentrations, continued enhancement of emission control measures, technological innovations, and policy enforcement will be essential. These efforts would be crucial not only for improving urban air quality but also for mitigating associated climate impacts.

**Author contributions.** YZ initiated and designed the study and developed the statistical methodology. YZ, QM, and JF developed the model code. QM and JF performed the simulations and analysis. YZ, SZ, YR, JQ, LT, and MZ conducted field measurements and validated the data. QM and YZ prepared the original manuscript, and XX, JP, OF, and XG provided comments on the manuscript.

**Competing interests.** The contact author has declared that none of the authors has any competing interests.

**Acknowledgments**. This study was supported by the National Natural Science Foundation of China (grant no. 42207124) and Natural Science Foundation of Jiangsu Province (grant no. BK20210663).

**Data availability.** The MERRA-2 (M2T1NXADG, V5.12.4) data can be downloaded at https://goldsmr4.gesdisc.eosdis.nasa.gov/data/MERRA2/M2T1NXAER.5.12.4/.
The ERA5 reanalysis data can be downloaded at https://cds.climate.copernicus.eu/cdsapp#!/dataset/reanalysis-era5-pressure-levels?tab=overview.
The TAP data is from http://tapdata.org.cn/ (Geng et al., 2021). The additional data will be made available upon request (yjzhang@nuist.edu.cn).

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
