# Peer review of "Technical note: Reconstructing surface missing aerosol elemental carbon data in long-term series with ensemble learning"

_EGUsphere, 2024_

## Referee Comment (RC2)

The authors adopt one ensemble learning model by integrating three Machine Learning models, including Gradient Boosting Regression Trees (GBT), eXtreme Gradient Boosting (XGB) and Random Forest (RF), coupled with ridge regression to generate robust predictions, to fill the gap of the element carbon (EC) data from 2013 to 2023 in Yangtze River Delta, China. The reconstructed EC dataset is valid by the intercomparison of EC with other datasets. Lastly, ensemble learning was used to design a fixed emission approximation method to disentangle and quantify the contribution of anthropogenic drivers to EC reduction.

This work is well organized. The authors present sufficient evidence to prove their robust and good performance in terms of the ensemble learning method. However, I'm sceptical about certain results of this study, particularly on the fixed emission approximation method. The acceptance of this manuscript is contingent upon the authors thoroughly validating those results. In addition, several places in this manuscript require an improvement. I recommend the acceptance after the authors address the comments and concerns detailed below.

**General comment:**

After reading this manuscript, my initial impression is that the authors have a wide knowledge of Machine Learning. However, I have some concerns as follows: As you mentioned in the 2.4.3 section (**Line: 225**): the errors increase when 2018 and 2019 are used as baseline years. 1) I am confused by the reason you provided, which is due to the missing meteorological parameters. As far as I know, ERA5 is a continuously updated dataset. It should not have missing values in 2018 and 2019. Please clarify this point. 2) If possible, try to use the ground-based measurements of meteorological factors rather than ERA5; 3) Please clarify how you retrieved the meteorological factors from the ERA5 in four cities in the 2.1 section. 4) In principle, the choice of the baseline year is critical. Basically, the baseline year is representative of typical conditions. If the selected year is an anomaly (e.g. huge emission reduction in COVID year), it could lead to an overestimation/underestimation. Could you explain how you chose the baseline year?

**Specific comment:**

1) **Line 27**: Rephase the sentence: from 2013 to 2020 (-0.24 to -0.15 µg m$^{-3}$ a$^{-1}$) from 3.26 µg m$^{-3}$ to 1.59 µg m$^{-3}$

2) When narrating, maintain consistency in sentence tenses. For example, we evaluated… in Line 199 and we propose… in Line 206

3)**Line 214:** If the FEA method were…. Please double-check the whole text and use the singular and plural correctly.

---

## Author Comment (AC1)

**Reply to Referee # 2**

Long-term in-situ observations of black carbon aerosols are crucial for studying their environmental and climatic effects. However, in real-world observational studies, there are several inevitable technical challenges, such as data gaps. This paper proposes a machine learning method that elegantly addresses this issue. The method is applied to reconstruct time-series data of elemental carbon (EC) aerosols from four cities in eastern China. The results are also validated by comparing them with other datasets. Furthermore, the paper introduces a novel method for assessing the driving factors of long-term trends in elemental carbon, as well as evaluating the uncertainty associated with this approach. I believe both methods hold significant value for the field of atmospheric monitoring. Overall, the paper is well designed and written. However, I have the following points that the authors should address:

**Reply**: We would like to express our gratitude to the reviewer for the positive and constructive comments. We have carefully revised the manuscript in accordance with the suggestions provided. Please find our responses below:

The authors introduce MERRA-2 black carbon column concentration data as one of the predictor variables. They also compare MERRA-2 near-surface black carbon concentrations and find that the MERRA-2 data tends to overestimate the site's elemental carbon data. I suggest that the authors conduct a sensitivity test by training the machine learning model without using MERRA-2 black carbon column concentration as a predictor variable and compare the results with the current ones.

**Reply**: Thank you for your suggestion. We have conducted a sensitivity test by training the machine learning model without including MERRA-2 black carbon column concentration (BCC) as a predictor variable. We compared the results with those of the current model and present the findings in Figure S4. The revised text reads: "*We further evaluated the importance of the MERRA-2 black carbon concentration (BCC) as a predictor by testing the model's performance both with and without this variable (see Figure S4). Inclusion of MERRA-2 BCC significantly improved the model's performance across all evaluation metrics, confirming it as a key contributor to model accuracy.*" More detailed modifications can be found on lines 206 - 209 in the revised manuscript.

[Figure]

**Figure S4.** Comparison of the model performance parameters for four cities, training the machine learning model with or without MERRA-2 BCC as a predictor variable (The darker color represents with MERRA-2, while the lighter color represents without MERRA-2.).

The trend changes in EC aerosols are influenced by both meteorological conditions and emissions. In eastern China, the sources of black carbon generally include vehicle emissions and industrial coal combustion. While the paper quantifies the overall anthropogenic emission trend drivers, there is relatively little information on specific emission sectors, which may be a limitation of the method employed. The paper analyzes the daily variation of EC over the years and suggests that the reduction of motor vehicle emissions may be a major factor driving the decline in EC levels. I suggest that the authors could try to extend this analysis by investigating the trend changes of EC during vehicle emission rush hours or by quantifying the driving factors for these peak periods. This could provide a more detailed understanding of the trend changes.

**Reply**: We appreciate the suggestion to further explore the trend changes in EC during vehicle emission rush hours. We have extended the analysis to include this investigation and quantified the driving factors for these peak periods, which has provided a more detailed understanding of the trend changes. The results are shown in Figures S14 and S15.

[Figure]

**Figure S14.** Trends in monthly average EC concentrations during morning rush hours (07:00–09:00) from 2013 to 2023 across the four cities: Nanjing, Suzhou, Xuzhou, and Zhenjiang.

[Figure]

**Figure S15.** Drivers of EC trends from 2013 to 2023. (**a–d**) Contributions of anthropogenic emission control and meteorological variations to EC concentration trends across four cities. (**e–h**) Same as (**a–d**), but specifically for rush-hour periods (7:00 – 9:00) in the four cities.

The revised text reads: "*Figure S6a–d and Table S7 further demonstrate strong correlations between EC and NO₂ concentrations, suggesting that vehicle emissions are a major contributor to urban BC levels. The observed reductions in NO₂ in all four cities underscore the impact of transportation-related emission control measures on declining EC concentrations. We further examined rush-hour EC trends in Figures S14 and S15 and found more pronounced declines during these periods compared to the interannual average, indicating that targeted control strategies for mobile sources have been particularly effective. This finding aligns with previous work (e.g., Zheng et al., 2018), which reported substantial reductions in BC emissions from the transportation sector in China between 2010 and 2017.*" More detailed modifications can be found on lines 372 - 378 in the revised manuscript.

The authors use the ridge regression algorithm for the multivariate regression analysis but do not employ the traditional multiple linear regression algorithm. I recommend that the authors clarify this choice. Additionally, regarding Equation 1, the expression may cause confusion because GBRTs, XGBoost, and RF are abbreviations for different machine learning algorithms, yet they are presented as variables in the formula. I suggest the authors optimize the notation for clarity.

**Reply**: Thank you for the comment. We chose ridge regression for multivariate regression analysis due to its ability to handle multicollinearity, which is often present in complex models. Ridge regression introduces a regularization term that enhances model stability and reduces the risk of overfitting. This approach ensures more reliable estimates when dealing with correlated predictors. The revised text reads: "*For multivariate regression analysis, we chose ridge regression over traditional multiple linear regression to account for multicollinearity among the three model outputs. Ridge regression is particularly effective in handling multicollinearity by introducing a regularization term that improves computational stability and reduces the risk of overfitting (Kidwell and Brown, 1982; Hoerl and Kennard, 1970).*"

We have also revised the notation to clarify that GBRTs, XGBoost, and RF are machine learning algorithms, not variables, to avoid any confusion. The revised equation reads: "

$$f_{EL} = m_1 f_{GBRTs} + m_2 f_{XGBoost} + m_3 f_{RF}, \qquad\qquad\qquad (1)"$$

More detailed modifications can be found on lines 130 – 133 and 135 in the revised manuscript.

Line 148 – 149: Appropriate references should be cited to support the use of these pollutants as tracers for source characterization.

**Reply**: Thank you for the comment. We have added references to support the use of these pollutants as tracers for source characterization. More detailed modifications can be found on lines 142 - 143 in the revised manuscript.

Line 239: The phrase "Reconstruction of missing data of EC and trend analysis" should be revised to "Reconstruction of missing data of EC and comparison".

**Reply**: Thanks for the suggestion. Modified.

Line 336 – 337: The discussion on the impact of COVID-19 lockdowns on EC trend changes is well noted as a factual observation. Could the authors further discuss or quantify such impact?

**Reply**: Thank you for your valuable suggestion. We have further discussed the impact of the COVID-19 lockdowns and quantified their contribution to EC trend changes. Our analysis, as shown in Figure 5, indicates that the COVID-19 lockdown contributed approximately 3%-10% to the total reduction in EC concentrations between 2017 and 2020, taking into account both meteorological variations and changes in anthropogenic activities.

[Figure]

**Figure 5.** Drivers of the EC trend from 2013 to 2023. a-d the contributions of anthropogenic emission control, meteorological conditions and COVID-19 lockdown on the trends in EC concentration in the four cities.

The revised text now reads: "*As shown in Figure 5, we quantified the impact of the lockdown on EC levels, estimating that the pandemic contributed 3–10% of the total EC reduction from 2017 to 2020—an effect stemming from both emission reductions and changes in human activity. This is consistent with Zheng et al. (2021), who reported large-scale BC emission reductions during the lockdowns of early 2020, especially from industrial, residential, and transportation sources.*" More detailed modifications can be found on lines 181 – 190 and 398 – 402 in the revised manuscript.

---

## Author Comment (AC2)

**Reply to Referee # 3**

The authors adopt one ensemble learning model by integrating three Machine Learning models, including Gradient Boosting Regression Trees (GBT), eXtreme Gradient Boosting (XGB) and Random Forest (RF), coupled with ridge regression to generate robust predictions, to fill the gap of the element carbon (EC) data from 2013 to 2023 in Yangtze River Delta, China. The reconstructed EC dataset is valid by the intercomparison of EC with other datasets. Lastly, ensemble learning was used to design a fixed emission approximation method to disentangle and quantify the contribution of anthropogenic drivers to EC reduction. This work is well organized. The authors present sufficient evidence to prove their robust and good performance in terms of the ensemble learning method. However, I'm sceptical about certain results of this study, particularly on the fixed emission approximation method. The acceptance of this manuscript is contingent upon the authors thoroughly validating those results. In addition, several places in this manuscript require an improvement. I recommend the acceptance after the authors address the comments and concerns detailed below.

**Reply:** The authors thank the referee for their thoughtful and constructive comments. We have made the necessary revisions to address the concerns and suggestions raised. Below is a summary of the changes made in response to the referee's comments:

General comment:

After reading this manuscript, my initial impression is that the authors have a wide knowledge of Machine Learning. However, I have some concerns as follows: As you mentioned in the 2.4.3 section (Line: 225): the errors increase when 2018 and 2019 are used as baseline years. 1) I am confused by the reason you provided, which is due to the missing meteorological parameters. As far as I know, ERA5 is a continuously updated dataset. It should not have missing values in 2018 and 2019. Please clarify this point.

**Reply**: Sorry for the confusion in our initial explanation. The higher uncertainties observed in those years stem from using different baseline years in the Fixed Emission Approximation (FEA) method. These uncertainties reflect the variability in results when different years are used as baselines. To address this, we have revised the manuscript to remove the original explanation and have instead provided a clearer presentation of the uncertainty range in the results (see Figure 5).

[Figure]

**Figure 5.** Drivers of the EC trend from 2013 to 2023. a-d the contributions of anthropogenic emission control, meteorological variations and COVID-19 lockdown on the trends in EC concentration in the four cities.

2) If possible, try to use the ground-based measurements of meteorological factors rather than ERA5;

**Reply**: Thank you for referee's suggestion. While we agree that ground-based measurements are valuable, we explain the challenges associated with their use in this study. Ground-based measurements often do not provide the full set of 18 meteorological parameters offered by ERA5, and they can also be affected by missing values. We will continue to explore ways to integrate such measurements when feasible.

3) Please clarify how you retrieved the meteorological factors from the ERA5 in four cities in the 2.1 section.

**Reply**: Thank you for referee's suggestion. We clarified how we retrieved the meteorological factors from ERA5 for the four cities in Section 2.1. The revised text now reads: "*To represent the meteorological conditions at the observation sites, we extracted data from the ERA5 grid cells that correspond to the coordinates of the monitoring stations.*" More detailed modifications can be found on lines 90 - 91 in the revised manuscript.

4) In principle, the choice of the baseline year is critical. Basically, the baseline year is representative of typical conditions. If the selected year is an anomaly (e.g. huge emission reduction in COVID year), it could lead to an overestimation/underestimation. Could you explain how you chose the baseline year?

**Reply**: Thank you for referee's suggestion and comments. This is a very good point. We addressed the concern about baseline year selection by explaining that we considered multiple potential baseline years and averaged the results for each year. This helped mitigate any anomalies, such as those caused by the emission reductions during the COVID-19 lockdown. Error bounds were also provided in Figure 5 to reflect the range of potential baseline year impacts.

Specific comment:

1)Line 27: Rephase the sentence: from 2013 to 2020 (-0.24 to -0.15 $\mu g\ m^{-3}\ a^{-1}$) from 3.26 $\mu g\ m^{-3}$ to 1.59 $\mu g\ m^{-3}$

**Reply**: Thanks for the suggestion. The sentence indeed had some ambiguity. The revised text now reads: "*Over the 11-year period, EC exhibited an overall decline (-0.20 to -0.14 $\mu g\ m^{-3}\ a^{-1}$), with a more significant decrease from 2013 to 2020 (-0.24 to -0.15 $\mu g\ m^{-3}\ a^{-1}$). During this time, the average EC concentration in the four cities dropped from 3.26 $\mu g\ m^{-3}$ to 1.59 $\mu g\ m^{-3}$, followed by a noticeable slowdown in the rate of decline from 2020 to 2023 (-0.12 to -0.04 $\mu g\ m^{-3}\ a^{-1}$).*" More detailed modifications can be found on lines 25 - 28 in the revised manuscript.

2) When narrating, maintain consistency in sentence tenses. For example, we evaluated… in Line 199 and we propose… in Line 206

**Reply**: Thanks for the suggestion. We agree that consistency in sentence tenses is important. We have used the past tense throughout, and the necessary changes have already been made.

3)Line 214: If the FEA method were…. Please double-check the whole text and use the singular and plural correctly.

**Reply**: Thanks for the suggestion. Modified. The entire text has been reviewed to ensure correct usage of singular and plural forms.

---

## Author Comment (AC3)

**Reply to Referee # 4**

The current manuscript aims to address the lack of continuous data for 4 cities gap-filled black carbon (BC) data to ultimately assess the trends in this pollutant as a result of the mitigation plans enforced in China in the 2013-2023 period. The reconstruction of these measurements is conducted by means of a machine learning (ML) ensemble of techniques validated upon the existent data, providing good agreement for all sites and years. Additionally, this manuscript provides a method to estimate weather and emission contributions to the reported concentrations, hence tackling the assessment of the effectivity of the abatement actions based solely on the anthropic drivers of BC. The reviewer agrees to publish this article under minor revisions.

**Reply:** The authors sincerely thank the reviewer for their thoughtful and constructive comments. We have carefully addressed the suggestions and made the necessary revisions to improve the manuscript.

Overall Feedback: The presented manuscript is outstanding regarding the implementation of machine learning in atmospheric aerosol studies while maintaining the final purpose of it, evaluating the trends of the studied pollutant as a consequence of the implemented abatement plans. This paper consists on three main blocks: i. Gap filling of BC time series; ii. Differentiation of the anthropogenic and meteorological drivers of BC evolution; iii. Trend analysis of the outcoming i., ii., outcomes to evaluate China's pollution mitigation actions. The manuscript is in general very well-written and structured. However, I list below certain aspects which should be addressed:

**Reply:** The authors sincerely thank the reviewer for their constructive suggestions and comments. We have carefully addressed each of the suggestions and comments, and the detailed responses can be found below:

The BC, EC data used in the EL models are not clearly described, neither the conversion from one to the other. Firstly, please, state for which cities you have EC, for which you have rBC, and for which you have both (Nanjing only, I assume). I see how Figure 1 shows a 1:1 slope for the presented sites and the Nanjing dataset, but it can not be like this for every site (Jeong

et al., 2004, Rigler et al., 2020). Since you cannot provide a 1:1 EC-BC scatterplot for your other sites, please at least state the risk of EC, BC not being interchangeable in the rest of your sites and indicate possible consequences of that.

**Reply**: Sorry for the confusion. Long-term EC measurements were available for all four cities (i.e., Nanjing, Suzhou, Xuzhou, and Zhenjiang), while refractory black carbon (rBC) data, based on a short-term field campaign, were only available for Nanjing. In the revised manuscript, we have provided a clearer description of the data availability. We agree with the reviewer's point that the 1:1 relationship observed between EC and rBC in the original Figure 1 for Nanjing cannot be assumed to apply to the other cities. To improve clarity, we have made corresponding revisions in the manuscript and placed the revised figure in the supplementary materials as Figure S1. The revised text reads: "*In this study, long-term measurements of EC were conducted across four cities (i.e., Nanjing, Suzhou, Xuzhou, and Zhenjiang), using the Sunset Laboratory semi-continuous OC/EC analyzer (Model-4), which provides hourly time-resolution. ... In addition to EC measurements, refractory black carbon (rBC) data from our previous study in Nanjing (Yang et al., 2019) were used for inter-comparison with EC data. ... Figure S1 shows the relationship between rBC and EC mass concentrations for the Nanjing dataset in this study, along with those from previous work (Pileci et al., 2021). The results reveal a good agreement between rBC and EC (slope = 1.01) in Nanjing, which is consistent with the range reported by Pileci et al. (2021).*" More detailed modifications can be found on lines 93 – 94, 97 – 99 and 103 – 105 in the revised manuscript.

You mention in 2.2, 2.3 the limitations of measurements and simulations and the substantial uncertainty these could drag to the EL model. Did you consider introducing uncertainties of both measurements and models as predictors in your EL? In case they became a strong predictor, you could narrow down which instrumental errors are more problematic for your data reconstruction, and maybe you could improve the predictions if filtering them out.

**Reply**: Thank you for this insightful suggestion. We agree that uncertainties in both measurements and simulations can significantly affect the prediction accuracy of the ensemble learning model (EL). These uncertainties likely extend to key variables such as air pollutant gases and meteorological parameters used in the data reconstruction process. Introducing uncertainty estimates as predictors could indeed help identify which types of instrumental or

model-related errors most strongly influence the model performance and potentially allow for improvements by filtering or weighting inputs. However, due to the lack of comprehensive uncertainty quantification for all relevant variables across the study period and locations, integrating them into the current EL framework remains challenging. Nonetheless, we truly appreciate this constructive recommendation and will explore this direction in our future work to enhance model robustness and interpretability.

Line 134. Provide some explanation on the advantages of the ridge regression or a reference.

**Reply**: Thanks for the suggestion. We chose ridge regression for multivariate regression analysis due to its ability to handle multicollinearity, which is often present in complex models. Ridge regression introduces a regularization term that enhances model stability and reduces the risk of overfitting. This approach ensures more reliable estimates when dealing with correlated predictors. The revised text reads: "*For multivariate regression analysis, we chose ridge regression over traditional multiple linear regression to account for multicollinearity among the three model outputs. Ridge regression is particularly effective in handling multicollinearity by introducing a regularization term that improves computational stability and reduces the risk of overfitting (Kidwell and Brown, 1982; Hoerl and Kennard, 1970).*" More detailed modifications can be found on lines 130 – 133 in the revised manuscript.

Please provide a list of all the "meteorological and emission indicator variables" (Line 151) that you feed the model with.

**Reply**: Thank you for your suggestion. As shown in Table S2, we have provided the list of meteorological and emission indicator variables that are fed into the model.

**Table S2.** Comparison Table of Meteorological and Emission indicator Variables

| Variable abbreviations | Meteorological and Emission indicator variables | Unit |
|---|---|---|
| U10 | 10m u-component of wind | $m\ s^{-1}$ |
| V10 | 10m v-component of wind | $m\ s^{-1}$ |
| U850 | 850hPa u-component of wind | $m\ s^{-1}$ |
| V850 | 850hPa v-component of wind | $m\ s^{-1}$ |

| | | |
|---|---|---|
| W850 | 850hPa w-component of wind | m s$^{-1}$ |
| U650 | 650hPa u-component of wind | m s$^{-1}$ |
| V650 | 650hPa v-component of wind | m s$^{-1}$ |
| W650 | 650hPa w-component of wind | m s$^{-1}$ |
| U500 | 500hPa u-component of wind | m s$^{-1}$ |
| V500 | 500hPa v-component of wind | m s$^{-1}$ |
| W500 | 500hPa w-component of wind | m s$^{-1}$ |
| Tmx | Maximum 2m temperature | K |
| BLH | Boundary layer height | m |
| RH | Relative Humidity | Dimensionless |
| SR | Mean surface direct short-wave radiation flux | W m$^{-2}$ |
| SP | Mean sea level pressure | Pa |
| TCC | Total cloud cover | Dimensionless |
| TP | Total precipitation | m |
| CO | Carbon Monoxide | mg m$^{-3}$ |
| SO$_2$ | Sulfur Dioxide | μg m$^{-3}$ |
| NO$_2$ | Nitrogen Dioxide | μg m$^{-3}$ |
| BCC | Black Carbon Column Mass Density | μg m$^{-2}$ |

The proportion of data trained vs. reconstructed is concerning. Even if you get good reconstructive metrics, I feel a bit skeptical on how extrapolating these predictions learned to other years can be an oversimplification, especially if the years to be reconstructed are anterior to the mitigation policies, as for Xuzhou, Zhenjiang. You could be missing actual significant drivers of EC that were minimized after the abatement regulations. Please, consider evaluating such long-term trends for these two last cities if you don't have any measurements/satellite information about the previous atmospheric composition. Also, provide the correlation with CO, NO$_x$ you gave for the whole period only in the reconstructed periods in addition to the overall long-term correlation.

**Reply**: Thank you for your thoughtful suggestions and insightful comments. We agree with the reviewer that the potential oversimplification in extrapolating predictions to years preceding the implementation of mitigation policies, as this could overlook significant drivers of EC that

were reduced after regulatory measures. To better account for emission influences, we incorporated emission indicators, including ground-based observations of three conventional air pollutants (CO, $NO_2$, and $SO_2$), as well as MERRA-2 BCC data, during the reconstruction period. To assess the model's robustness, we also evaluated its performance without these emission indicators. As the reviewer correctly pointed out, the model performance for extrapolated years significantly declined without these variables. This finding, presented in Figure S5, highlights the importance of incorporating emission data to ensure reliable predictions. In addition, we have now provided the correlation between CO and $NO_2$ specifically for the reconstructed periods, alongside the overall long-term correlation, as shown in Figure S6. The CO/$NO_2$ ratio during the observational period closely matches that during the EC-missing period, suggesting a limited impact of year-round variations in emission indicator ratios on model predictions. Furthermore, we conducted a comparative analysis between TAP BC data and both the EC-missing and EC-observed periods (Figures S10 and 3). The results demonstrate a good agreement between these two periods, further validating the reliability and acceptability of the reconstructed data. Nevertheless, we have expanded our discussion for the potential uncertainties in historical data reconstruction due to emission reductions. More detailed modifications can be found on lines $210 - 215$ in the revised manuscript. We appreciate the reviewer's valuable input to enhance the robustness of these findings.

[Figure]

**Figure S5.** Comparison of the model performance parameters for four cities, training the machine learning model with or without emission indicators (EI) as a predictor variable (The darker color represents with emission indicators, while the lighter color represents without emission indicators.).

[Figure]

**Figure S6.** Correlation between CO and NO$_2$ concentrations. Panels (a–d, in blue) represent periods with available ground-based EC observations, while panels (e–h, in red) correspond to periods with missing EC observations.

Figure 4. Could be the comparison between EC (EL predictions), BCC (MERRA-2), and BC (TAP) misleading the interpretation of the plots here since these are not directly exchangeable variables? Please discuss the limits of the comparability.

**Reply**: Thank you for your comment. Yes, it is crucial to discuss the limitations of comparability among these datasets. The black carbon (BC) data from the TAP dataset are derived from an atmospheric chemical transport model. As a result, the simulated BC concentrations in TAP were more representative of EC concentrations. Our comparison indicates a good agreement between these datasets, suggesting the TAP BC dataset was more representative of EC. On the other hand, the BC concentrations in MERRA-2 were estimated by assimilating satellite-derived aerosol optical depth (AOD) into an atmospheric chemical transport model (Gelaro et al., 2017). Since satellite retrievals were based on optical properties, the inferred BC concentrations may be affected by optical effects, such as the lensing effect of coated BC particles, potentially leading to an overestimation of BC concentrations. Additionally, uncertainties in the aerosol-type classifications within the atmospheric chemical model could introduce further biases. These factors may partly explain why MERRA-2 BC concentrations tend to be higher than those in the TAP dataset, as well as the machine-learning-reconstructed or observed EC concentrations. We have incorporated these discussions into the revised manuscript to clarify the limitations and potential biases in comparing these datasets.

The revised text reads: "*The BC concentrations in MERRA-2 are estimated by assimilating satellite-derived aerosol optical depth (AOD) into an atmospheric chemical transport model (Gelaro et al., 2017). Since satellite retrievals are based on optical properties, the inferred BC concentrations be affected by optical effects, such as "lensing effect" (Liu et al., 2015), which could potentially lead to an overestimation of BC concentrations. ... For simplicity, TAP BC did not account for methodological differences between BC and EC measurements (Liu et al., 2022).*". More detailed modifications can be found on lines 110 – 113 and 117 – 118 in the revised manuscript.

I see the FEA method power to discern between meteorological and anthropic emissions, I consider this is a very well-conceived approach. However, I would restrict the *is* to be quite near the *js, ks*. Training with 2013 and predicting 2022, 2023 might be unrealistic, since the validity of the fixed emission hypothesis is less robust. This is specially concerning when training is performed with the reconstructed data with no measurements to validate these years, as I mentioned two points ago. I think being conservative here and acknowledging the limitations of your datasets would make your trend evaluations more sturdy, especially since some readers might be rather ML-skeptic.

**Reply**: Thank you for your insightful comments. We incorporated available observational data in place of the reconstructed data and applied the FEA method to analyze the driving factors in Nanjing and Suzhou, where a relatively large volume of observational data was available. The analysis revealed that the driving factor results obtained using observational data were generally consistent with those derived from the fully reconstructed dataset, as shown in Figure R1. These results validated the applicability of the reconstructed data for the FEA method.

[Figure]

Figure R1. Drivers of the reconstructed (RE) EC and observed (OBS) or RE EC trend from 2013 to 2023. The contributions of anthropogenic emission control and meteorological variations on the trends in EC concentration in the cities. OBS or RE, it is a combination of observed EC and reconstructed EC, with only the missing observed values being replaced by the reconstructed data.

Additionally, we tested the differences in FEA analysis results using reconstructed data obtained from models trained with varying amounts of observational data. As shown in Figure S7, the differences between the two datasets exist but remain within approximately 10% overall. Furthermore, to further validate the robustness of the FEA method, we utilized $PM_{2.5}$ data from the TAP dataset to ensure comparability with the approach of Zhang et al. (2019a), who employed a traditional atmospheric chemical transport model with a fixed emission inventory. As shown in Figure R2, the difference between our FEA-derived results and the simulation results of Zhang et al. (2019a), particularly in terms of emission factors, was 11%.

In summary, the reviewer has provided valuable suggestions, and we have accordingly revised the manuscript. The revised manuscript reads as: "*Additionally, since our study incorporates reconstructed data, discrepancies between the modeling training data and observational data may introduce additional uncertainties. To evaluate this, we selected Nanjing as a test case. Specifically, we reconstructed the 2013–2023 dataset using training data from two different periods: 2013–2020 and 2014–2019. We then applied the FEA method to analyze the results and assess their differences. As shown in Figure S7, while some variations exist between the two datasets, the overall difference remains within approximately 10% on average. This suggests that the choice of training datasets introduces a degree of uncertainty in the FEA results, which is inherent to machine learning and statistical modeling-based approaches.*". The modifications can be found on lines 255 – 261.

[Figure]

Figure S7. Cross-matrix relative difference analysis of the FEA method for Nanjing. (a) Relative difference in FEA analysis results based on the dataset reconstructed using observational data from 2013 to 2020. (b) Relative difference in FEA analysis results based on the dataset reconstructed using observational data from 2014 to 2019.

[Figure]

Figure R2. The contributions to EC concentration changes driven by anthropogenic emission control versus meteorological conditions(a) WRF-CMAQ model (Zhang et al., 2019a) and (b) FEA analyses during the 2013-2017 period.

Please indicate explicitly that $C_{MET(i,i)}$ is the self-prediction for the year i based on the training i. This can be understood from the text but stating it would help the reader to understand more easily since these nomenclatures might be new for them.

**Reply**: Thanks for the suggestion. This modification will indeed help the readers understand more easily. The revised text reads: "*The term $C_{MET(i,i)}$ is the self-prediction for the year i based on the training year i.*" More detailed modifications can be found on lines 227 – 228 in the revised manuscript.

In the text (lines 222-223), you provide uncertainties for these Ys. Can you please explain how did you get those uncertainties?

**Reply**: Thank you for your suggestion. The expression may not have been sufficiently precise. Instead of referring to these values as "uncertainties," we should clarify that they represent the "Relative difference (%)". This relative difference was calculated using Eqs. (11) and (12), and the values provided in the text correspond to the statistical evaluation of the relative differences derived from the model predictions for a specific year.

About Figure 3, how do you explain that the uncertainties of your methods are for almost all cells positive? If I understood properly, the negative uncertainties should be as probable as the positive ones, since $|\Delta ANT_{i,j}| \sim |\Delta ANT_{j,i}|$.

**Reply**: Yes, the reviewer is correct that negative uncertainties should, in principle, be as probable as positive ones, as both represent the relative deviation of the model. However, in our case, we observed more positive values. This asymmetry may arise due to systematic biases in the model predictions, the nature of the data distribution, or the specific characteristics of the reconstruction process. To ensure clarity and avoid potential misinterpretation, we have revised Eqs. (11) and (12) accordingly. More detailed modifications can be found on lines 238 and 242 in the revised manuscript. Furthermore, our newly proposed FEA method requires further investigation and validation, particularly in analyzing trends of different particulate matter chemical components. In future research, we will continue to explore these uncertainties to refine and improve the method.

About Figure 3, the fact that the lower uncertainties you get are from Xuzhou, with less measurements availability, whilts Suzhou, with higher coverage has higher uncertainty. This, for me, is reinforcing the idea that predicting over no measurement-anchors in the Xuzhou, Zhenjiang early period can lead to an oversimplification of the BC concentrations which might be comfortable for the FEA method. I find more normal that the model struggles for the actual measurement-based 2018-2019 baseline periods than that it doesn't for the predicted 2013-2015.

**Reply**: That's an excellent point, and we appreciate the reviewer's perspective, which provides new insights into the FEA method. The choice of different years as training data can indeed

introduce uncertainties. To account for this, we applied a cross-validation approach across multiple years. This not only allows us to quantify the distribution of uncertainties but also helps diagnose which years contribute the most to overall uncertainty in the modeling process. As the reviewer pointed out, the larger uncertainties in certain cities may stem from residual discrepancies between the reconstructed and observed data, which can amplify the uncertainties within the FEA method. This consideration has been incorporated into our discussion to better address potential limitations.

We have incorporated this consideration into our discussion to better address potential limitations of the method. Specifically, we have expanded the discussion on uncertainties arising from different training datasets in the FEA approach. The revised text now states: "*Because this study incorporates reconstructed EC data, potential discrepancies between model-derived training data and actual observations may introduce additional uncertainty into the results. To assess this, we selected Nanjing as a representative test case. Specifically, we reconstructed the 2013–2023 EC dataset using training data from two distinct time periods: 2013–2020 and 2014–2019. We then applied the FEA method to both reconstructions and compared the outcomes. As shown in Figure S7, while some variations exist between the two sets of results, the average difference remains within approximately 10%. This finding suggests that the choice of training dataset can introduce a moderate degree of uncertainty to the FEA results—an inherent characteristic of ensemble learning and other statistical modeling approaches. Nonetheless, the relatively small magnitude of this difference reinforces the reliability and generalizability of the machine-learning-based FEA framework.*" More detailed modifications can be found on lines 250 – 257 in the revised manuscript.

Please provide the trend-estimator method you use in the methods section (is it Senn's slope) and provide the significance estimator of your results. Do you use "seasonal" Senn slopes, so that their effect is less taken into account?

**Reply**: Sorry for confusion. We performed the trend estimation using the Mann-Kendall test combined with the seasonal Theil-Sen estimator. This approach accounts for seasonal variations and reduces their influence on trend detection. We have now explicitly stated this in the methods section. The revised text reads: "*Finally, long-term trends in monthly mean EC or BC concentrations were assessed using the non-parametric Mann–Kendall (MK) trend test. To*

*account for seasonal variability, trend slopes were derived using the seasonal Theil–Sen estimator, enhancing robustness in the presence of periodic fluctuations.*" Further details and modifications can be found on lines 215 – 218 in the revised manuscript.

The discussion on why MERRA-2 is not properly capturing trends is very interesting (lines 244-251). Please, could you further detail which (meteorological/emissions) situations are better/worse captured by MERRA and TAP?

**Reply**: Thanks for the comments. To further elaborate, we focused on analyzing the Black Carbon Surface concentration (BCS) in MERRA-2 and the FEA BCS derived from meteorological-driven changes using the FEA method. In this approach, we utilized only the 2013 dataset as the training data and applied this model along with meteorological data from 2013 to 2023 to predict concentrations for the same period. Consequently, the predicted results can be considered as driven by emissions fixed to the 2013 level, with annual variations in meteorological conditions influencing the concentration changes. As shown in Figure S8, we observed good correlations between the MERRA BCS and FEA BCS, indicating that the MERRA-2 Black Carbon concentration trends align well with meteorologically-driven concentration changes. However, the influence of emission reductions on MERRA-2 concentrations remains relatively small, which is why MERRA-2 does not capture the trend changes over time. In contrast, the TAP BC concentration data shows clear trend changes, which are generally consistent with the changes observed in our reconstructed data (see Figure 3). This suggests that the TAP data may be more sensitive to emission reductions and better reflects their impact on BC concentration trends compared to MERRA-2. This difference can be attributed to the fact that TAP uses a more regionally-specific model, incorporating local emission inventories and pollution control measures, which may be more adept at capturing short-term fluctuations and reductions in emissions.

We have added such discussion on the revised manuscript, it now reads as: "*To investigate this inconsistency, we applied the FEA method to isolate the meteorological contributions to the observed trends. As shown in Figure S8, MERRA-2 BC trends exhibit good agreement with the FEA-derived BC values, suggesting that the interannual variability in MERRA-2 BC was largely governed by meteorological factors. In contrast, the TAP BC dataset shows clear downward trends that are more closely aligned with the changes observed in our reconstructed*

*EC data (Figures 3b and 3c). This indicated that the TAP dataset for the two cities was more sensitive to anthropogenic emission reductions and better captures their influence on BC concentration trends in recent years for the two cities. Additionally, MERRA-2 BC consistently overestimates EC concentrations compared to both ground-based measurements and reconstructed data. This overestimation has also been reported in previous studies (e.g., Xu et al., 2020), which documented substantial overpredictions of MERRA-2 BC in urban areas in the Yangtze River Delta of Eastern China, such as Shanghai and Hangzhou.*" Further details and modifications can be found on lines 269 – 275 in the revised manuscript.

[Figure]

Figure S8. Correlation analysis between the BCS data in MERRA2 and its BCS driven only by meteorological conditions. (a. Suzhou and b. Xuzhou). FEA BCS refers to the Black Carbon Surface concentration data driven by meteorological variations. The calculation method for FEA BCS is as follows: a model is trained using data from 2013, and then this model is applied along with meteorological data from 2013 to 2023 to predict the concentrations for the same period.

Table S5. Why do you think the Zhenjiang city reconstruction is significantly worse than the others.

**Reply**: Thank you for your comment. After re-evaluating the error metrics, including correlation coefficient (R) and slope in Table S4, we confirm that the reconstruction performance for Zhenjiang is comparable to that of other cities, with all regional R values exceeding 0.97. The perception of significantly worse performance might arise from insufficient visual emphasis on key metrics in the table. To improve clarity, we have highlighted

the relevant indicators in Table S4, particularly those comparing reconstructed data with ground-based observations, to enhance interpretability.

Please, provide a short explanation on the meteorological normalization method by Grange et al., 2018.

**Reply**: Thanks for the suggestion. The revised text reads: "*The normalization approach developed by Grange et al. (2018) adjusts pollutant concentrations by removing the influence of meteorological variability, thereby isolating the effects of emission control measures. Specifically, the method builds a statistical model to quantify the relationship between pollutant concentrations and meteorological and temporal variables. It then performs 1,000 resamplings of historical meteorological data while holding temporal variables constant at their observed values. The model calculates pollutant concentrations across all resampled meteorological scenarios, and the ensemble average of these predictions represents the meteorologically normalized (i.e., de-weathered) concentration. This process enables a more accurate attribution of observed trends to changes in emissions. Detailed methodology can be found in Grange et al. (2018) and related studies (Vu et al., 2019; Zhang et al., 2019b; Zhou et al., 2022).*" More detailed modifications can be found on lines 341 – 348 in the revised manuscript.

Figure S5e-h. It seems that ~2013, emission diels were rather flat whilst they become more marked in the last years. This could be because: i. Meteorological influence was underestimated for those periods; ii. Emission patterns/sources changed. Please discuss this variance.

**Reply**: Thank you for your comment. Sorry for any confusion caused by the previous labeling in the figure. The flatness observed around 2013 in Figure S11e-h is due to the methodology used. Specifically, the lines in the original figure represent data where the values for each year are subtracted from the baseline year of 2013. As a result, the emission trends appear relatively flat for the baseline period. To clarify, we have now updated the figure to use the label "RE-FE," as shown in Figure S11. This new label represents the difference between the reconstructed emissions (RE) and the fixed emissions (FE). This method is used to estimate emission changes relative to 2013, which is why emissions appear relatively stable around 2013. We hope this explanation resolves any uncertainties.

[Figure]

Figure S11. Diurnal variation of EC concentration. a–d. Diurnal variations of EC concentrations for each year from 2013 to 2023. e–h. Diurnal variations of EC concentrations for each year from 2013 to 2023 driven only by emission control. RE-FE represents the contribution of emission control to EC concentrations. It is calculated by training a model using data from 2013 and then applying this model to meteorological data from 2013 to 2023 to predict concentrations. The difference between the reconstructed concentrations and these predictions yields the RE-FE values.

In the last paragraph of your results section you explain the reductions of the anthropogenic emissions in the period of study, which is the objective of the paper. Could you also give some insights on the trend of meteorological impacts on concentrations? Do you consider that the atmospheric influence should be static over the trend or do you expect steady changes?

**Reply**: Thank you for the insightful suggestion. We have conducted additional analysis on the trend of meteorological impacts on EC concentrations. As shown in Figure S13, the meteorology-driven changes in EC concentrations, derived from both the FEA and deweather methods, remain relatively small. The annual contribution of meteorological effects is consistently below 0.23 µg m⁻³, suggesting that while meteorological variability plays a role in short-term fluctuations, its long-term impact on the overall trend is minimal compared to emission changes. Furthermore, our analysis indicates that meteorological contributions fluctuate over the 11-year period without a clear upward or downward trend. This suggests that

while the long-term impact of meteorology on EC concentrations remains relatively small, it is not entirely static and exhibits interannual variability. These fluctuations likely reflect changes in meteorological patterns, but they do not introduce a systematic bias in the overall trend of EC concentrations.

[Figure]

Figure S13. Trend in meteorological-driven EC from 2013 to 2023. Comparison of the results obtained from two different methods, including the FEA method developed in this study and the widely used de-weathered method. The EC concentrations driven only by meteorological conditions were calculated using the FEA method. In contrast, the de-weathered method first estimates EC concentrations influenced only by emission controls; by subtracting these values from the observed data, an approximation of the concentration changes driven by meteorological variability can be derived.

Technical changes

Figure 3 is a bit difficulty posed. I understand that the FEA uncertainty shown here is the Yi of equations 10, 11, please indicate this instead of "FEA uncertainty (%)" in the colourbar.

**Reply**: Thanks for the suggestion. We have updated "FEA uncertainty (%)" to the correct term "Relative difference (%)".

Figure 4, please play with the transparency or the wave order of the a-d time series so that we can see when the observations actually happen in a glance.

**Reply**: Thank you for the helpful suggestion. We have revised Figure 3 accordingly by adjusting the transparency and plotting order of the time series (a–d), so that the periods with actual observations can be more clearly identified at a glance.

[Figure]

**Figure R3.** Comparison of EC or BC concentration from different data sets. a-d Trends in EC or BC concentration from the four data sets (EL, MERRA-2, TAP, and observation). e-f Monthly variations. i-l Relationship of reconstructed, MERRA-2, and TAP modeled EC or BC with observed EC.

**References:**

Gelaro, R., McCarty, W., Suárez, M. J., Todling, R., Molod, A., Takacs, L., Randles, C. A., Darmenov, A., Bosilovich, M. G., and Reichle, R.: The modern-era retrospective analysis for research and applications, version 2 (MERRA-2), Journal of climate, 30, 5419-5454, 2017.

Grange, S. K., Carslaw, D. C., Lewis, A. C., Boleti, E., and Hueglin, C.: Random forest meteorological normalisation models for Swiss PM 10 trend analysis, Atmospheric Chemistry and Physics, 18, 6223-6239, 10.5194/acp-18-6223-2018, 2018.

Hoerl, A. E. and Kennard, R. W.: Ridge regression: Biased estimation for nonorthogonal problems, Technometrics, 12, 55-67, 1970.

Jeong, C. H., Hopke, P. K., Kim, E., & Lee, D. W. (2004). The comparison between thermal-optical transmittance elemental carbon and Aethalometer black carbon measured at multiple monitoring sites. Atmospheric Environment, 38(31), 5193-5204.

Kidwell, J. S. and Brown, L. H.: Ridge regression as a technique for analyzing models with multicollinearity, Journal of Marriage and the Family, 287-299, 1982.

Liu, S., Geng, G., Xiao, Q., Zheng, Y., Liu, X., Cheng, J., and Zhang, Q.: Tracking Daily Concentrations of PM2.5 Chemical Composition in China since 2000, Environmental Science & Technology, 56, 16517-16527, 10.1021/acs.est.2c06510, 2022.

Liu, S., Aiken, A. C., Gorkowski, K., Dubey, M. K., Cappa, C. D., Williams, L. R., Herndon, S. C., Massoli, P., Fortner, E. C., Chhabra, P. S., Brooks, W. A., Onasch, T. B., Jayne, J. T., Worsnop, D. R., China, S., Sharma, N., Mazzoleni, C., Xu, L., Ng, N. L., Liu, D., Allan, J. D., Lee, J. D., Fleming, Z. L., Mohr, C., Zotter, P., Szidat, S., and Prevot, A. S. H.: Enhanced light absorption by mixed source black and brown carbon particles in UK winter, Nature Communications, 6, 8435, 10.1038/ncomms9435, 2015.

Pileci, R. E., Modini, R. L., Bertò, M., Yuan, J., Corbin, J. C., Marinoni, A., Henzing, B., Moerman, M. M., Putaud, J. P., and Spindler, G.: Comparison of co-located refractory black carbon (rBC) and elemental carbon (EC) mass concentration measurements during field campaigns at several European sites, Atmospheric Measurement Techniques, 14, 1379-1403, 2021.

Rigler, M., Drinovec, L., Lavrič, G., Vlachou, A., Prévôt, A. S., Jaffrezo, J. L., ... & Močnik, G. (2020). The new instrument using a TC–BC (total carbon–black carbon) method for the online measurement of carbonaceous aerosols. Atmospheric Measurement Techniques, 13(8), 4333-4351.

Yang, Y., Xu, X., Zhang, Y., Zheng, S., Wang, L., Liu, D., Gustave, W., Jiang, L., Hua, Y., and Du, S.: Seasonal size distribution and mixing state of black carbon aerosols in a polluted urban environment of the Yangtze River Delta region, China, Science of The Total Environment, 654, 300-310, 2019.

Zhang, Q., Zheng, Y., Tong, D., Shao, M., Wang, S., Zhang, Y., Xu, X., Wang, J., He, H., Liu, W., Ding, Y., Lei, Y., Li, J., Wang, Z., Zhang, X., Wang, Y., Cheng, J., Liu, Y., Shi, Q., Yan, L., Geng, G., Hong, C., Li, M., Liu, F., Zheng, B., Cao, J., Ding, A., Gao, J., Fu, Q., Huo, J., Liu, B., Liu, Z., Yang, F., He, K., and Hao, J.: Drivers of improved PM2.5 air quality in China from 2013 to 2017, Proc. Natl. Acad. Sci. U.S.A., 116, 24463-24469, 10.1073/pnas.1907956116, 2019a.

Zhang, Y., Vu, T. V., Sun, J., He, J., Shen, X., Lin, W., Zhang, X., Zhong, J., Gao, W., and Wang, Y.: Significant changes in chemistry of fine particles in wintertime Beijing from 2007 to 2017: impact of clean air actions, Environmental Science & Technology, 54, 1344-1352, 10.1021/acs.est.9b04678, 2019b.